**Data Availability Statement:** All RNAseq files will be available from the SRA database upon publication, https://www.ncbi.nlm.nih.gov/sra,

**Funding:** This work was supported by National Science Foundation grant no. 1444987 to CDD and

# Identification of early fruit development reference genes in plum

**Kelsey Galimba[1], Roberta Tosetti[1¤a], Karen Loerich[2], Leann Michael[2], Savita Pabhakar[2¤b], Cynthia Dove[2], Chris Dardick[1], Ann Callahan[1]***

1 USDA-ARS, Appalachian Fruit Research Station, Kearneysville, United States of America, 2 Hagerstown Community College, Hagerstown, United States of America

¤a Current address: Plant Science Laboratory, Cranfield University, Bedfordshire, England, United Kingdom
¤b Current address: Department of Science, Frederick Community College, Frederick, United States of America
* ann.callahan@USDA.gov

## Abstract

An RNAseq study of early fruit development and stone development in plum, *Prunus domestica*, was mined to identify sets of genes that could be used to normalize expression studies in early fruit development. The expression values of genes previously identified from *Prunus* as reference genes were first extracted and found to vary considerably in endocarp tissue relative to whole fruit tissue. Nine other genes were chosen that varied less than 2-fold amongst the 20 RNAseq libraries of early fruit development and endocarp tissues. These gene were tested on a series of developmental plum fruit samples to determine if any could be used as a reference gene in the analyses of fruit-based tissues in plum. The three most stable genes as determined using RefFinder were *IPGD* (imidazole glycerol-phosphate dehydratase), *HAM1* (histone acetyltransferase) and *SNX1* (sorting nexin 1). These were further tested to analyze genes expressed differentially in endocarp tissue between normal and minimal endocarp cultivars. To determine the universality of those nine genes as fruit development reference genes, three other data sets of RNAseq from peach and apple were analyzed to determine the reference gene expression. Multiple genes exhibited tissue specific patterns of expression while one gene, the *SNX1*, emerged as possessing a universal pattern between the Rosaceae species, at all developmental stages, and tissue types tested. The results suggest that the use of existing RNAseq data to identify standard genes can provide stable reference genes for a specific tissues or experimental conditions under exploration.

## Introduction

One approach to select plants with improved traits is to identify the genes that affect that trait, negative or positively. Many studies employ RNA expression profiles to either identify or confirm that expression of a particular gene(s) correlates positively or negatively with the trait of interest. This would suggest that the genes could be involved with the trait. To avoid misleading interpretations about the correlation of the candidate gene with the trait, the expression of

AMC and by the United States Department of Agriculture—Agricultural Research Service to KDG, CDD and AMC. The funders had no role in study design, data collection and analysis, decision to publish, or preparation of the manuscript.

**Competing interests:** The authors have declared that no competing interests exist.

the gene of interest needs to be normalized. To normalize that gene expression, reference genes need to be analyzed in the same manner as the genes of interest [1–3]. Reference gene expression must not vary in the tissues, treatments or developmental times that are under study; moreover, the quality and amount of RNA would theoretically affect the reference gene in the same manner as the gene of interest.

There have been several published reports of reference genes for *Prunus*, initially developed using homologs to so-called housekeeping genes in other species [4]. It was clear that many of these genes in peach though were not expressed uniformly in all tissues and treatments. Further studies in peach and plum also demonstrated that the expression of these genes varied across tissues and treatments, so additional reference genes were identified [5–7]. Kim et al., [5] utilized RNAseq data for Japanese plum (*Prunus salicina*) flesh, peel and leaf tissues from two different cultivars at two different ripening stages to identify genes. They chose those with ρ >0.05 for differences between cultivars, >500 reads and with a coefficient of variation (CV) of <40%. Based on homology with peach sequences they selected 20 genes to test including the previous known standard reference genes [4]. The results of the analysis with reference gene programs showed that the best reference candidates were different for each experimental set. Overall and for S2 stage fruit expression, it was a combination of, a SAND-related trafficking protein MON (ppa003026), and an elongation factor 1 alpha, *EFIalpha* (ppa005702). The best standard genes for reproductive stage fruit tissue were a combination of the *MON* and an initiation factor (ppa012654). The authors concluded that each experimental set may need a very specific reference gene.

You et al. [6] utilized the genes in Kim et al. [5] as well as three candidate genes identified in plum peel, an 18S rRNA (based on TC1229); *CAC* (ppa006083); and *CATH* (ppa005912) to investigate the best reference genes combination during storage of plum fruit (*Prunus salicina*) at different temperature regimes. They concluded that for both room temperature and cold storage *CAC*, and *ACT*, and *UNK* (from Kim et al., [5]), were the optimal combination of reference genes. Similarly, Kou et al., [7] utilized RNAseq data from a peach fruit storage project to select the genes with the lowest CV across different temperature regimes. The overall conclusions of all these studies were that the reference genes are very dependent on the experimental set of RNAs being used. The three studies used various reference gene programs to predict the 'best' set of genes in a comparison for which all the programs predicted different 'best' genes. They concluded that a combination of reference genes based on the results of using RefFinder was one solution to picking the best genes for a stable profile.

We were interested in the mechanism for limiting endocarp production (stone tissue) in plum (*Prunus domestica*) [8–10]. To do so we analyzed a plum mutant, 'Stoneless' that developed only a small bit of stone or endocarp, apparently because of fewer cells in the endocarp layer. We performed an RNAseq experiment on a comparison of early fruit development and endocarp tissue between two normal cultivars and the 'Stoneless' cultivar (Tosetti *et al.* in preparation). In the process of confirming those results with qPCR, we found a need to develop reference genes appropriate for early fruit development and endocarp tissue in plum (*Prunus domestica*).

Therefore, we present the results of analyzing an RNAseq study in plum to determine which genes may represent good candidates as reference during early fruit development in plum. Our study included the previously identified *Prunus* reference genes and nine newly identified candidate reference genes that appeared to be more consistent in the RNAseq data from early fruit libraries. These were tested by qPCR on a series of developing plum fruit to determine the most stable. Their levels of expression were analyzed in additional RNAseq studies in peach and apple to determine how universal these reference genes might be for fruit studies.

## Materials and methods

### RNAseq data analysis

Plum RNAseq data (SRA# to be obtained) consisting of 20 libraries from whole ovary/whole fruit and endocarp tissue from two normal-stone cultivars ('Cacanska Lepotica' and 'Reine Claude de Bavay') and two individual trees from the 'Stoneless' cultivar ('Stoneless1' and 'Stoneless2') (S1 Table) at early timepoints was analyzed utilizing CLC Genomics Workbench (version 5.5) (Qiagen, Germantown, MD). Briefly, total RNA was submitted to David H. Murdock (Kannapolis, NC, USA) for sequencing via 100 base single end Illumina sequencing on a HiSeq 2000 and reads were then imported into CLC Genomics Workbench using standard import parameters. Reads were filtered by those matching ribosomal RNA (18S, 5.5S and 26S), mitochondrial RNA and chloroplast RNAs and the remaining reads were then mapped to the peach genome V1.0 [11; https://www.rosaceae.org]. Those that mapped were analyzed to count the reads via the RNAseq function of CLC. Because plum was being mapped to peach, the matches were set at 70% of the length of the read and 70% match. The counts were normalized (CLC Genomics WorkbenchVersion5.5) and unique read counts were exported to an Excel spreadsheet in order to be easily sorted. To predict gene functions, the transcript sequence was used to determine the best match to an *Arabidopsis* gene and its corresponding identity or function.

Peach and apple RNAseq data was analyzed in a similar manner, using a newer version of CLC Genomics Workbench (Version 11). In this case the unique reads per total mapped reads were directly obtained for the nine genes under analyses from the plum data sets. For apple, the peach sequence was used to BLAST the Apple Golden Delicious V1.0 [12; https://www.rosaceae.org) for the apple development series to find the best match. The two top matches were chosen as apple has a potentially doubled genome so that two genes might be expected. For the Apple hormone libraries, the nine candidate reference genes were used to BLAST the Apple Golden Delicious Haploid [13; https://www.rosaceae.org] Version 2.0 of the apple genome. The top two matches were used to obtain expression values from apple data sets.

### RNA extractions and qPCR

RNA was extracted from lyophilized plum tissue utilizing the Plant/Fungi Total RNA Purification Kit from Norgene Biotek (Thorold, ON Canada) following manufacturer's directions. RNA was DNased using a TURBO DNA-*free*™ Kit following manufacturer's directions (ThermoFisher Scientific, Waltham, MA). Following determination of quantity using a spectrophotometer, and RNA integrity on a 1.2% agarose gel, 600 ng was used to generate cDNA utilizing polyT tails and ProtoScript RT (New England Biolabs, Ipswich, MA). The resulting mix was diluted to 40 μl and 2 μl was used in a 20 μl PCR reaction with SsoAdvanced Universal SYBR Green Supermix by BioRad (Hercules, CA). Cycling conditions followed Roche Light Cycler 480 Real Time. Standard curves were run at the same time using one of the RNAs (the two earliest stages of the developmental series, DS1 or DS2, S3 Table) with fivefold dilutions.

### Analyses of qPCR

Cycle quantification values (Cqs) were taken from the automatic settings from the Light Cycler 480 software. Average values for the three technical reps were calculated. If one of the values was greater than 0.5 Cqs from the other two they were discarded. The standard curves were used to determine relative values by fitting the Cq value into the slope equation. Efficiency was calculated based on the slopes of the standard curves.

RefFinder [14] was used to compare the stability of the nine candidate reference genes. It was used on-line (http://150.216.56.64/referencegene.php?type=reference now maintained at https://www.heartcure.com.au/for-researchers/) and only took into account the raw Cq values without consideration of the efficiency of the reactions.

## Results

### Comparison of published *Prunus* reference genes (REF1-REF13)

Appropriate reference genes were needed to validate RNAseq expression profiles of early plum fruit development in a comparison between normal stone development and a mutant that developed considerably less stone. Previous studies on gene expression in peach and plum had used only the 26S rRNA gene as a reference gene [8–9]. Several studies since then presented candidate reference genes for *Prunus* [4–7]. The ten primer pairs from Tong et al., [4] were used to BLAST the peach V1.0 version of predicted transcripts. They matched to 16 predicted transcripts (Table 1). The number of unique plum reads that mapped to each of those

**Table 1. Expression values of peach reference genes[a] in RNAseq libraries in RPM.**

| # | Peach Transcript | *Arabidopsis* best match | avg RPMs | StDev RPM | High RPM | Low RPM | High/ Low RPM | CV (StDev/avg) |
|---|---|---|---|---|---|---|---|---|
| | **ACT7—Actin 7** | | | | | | | |
| REF1 | ppa007242 | AT5G09810.1 | 1,378 | 445.1 | 2,582 | 817.0 | 3.16 | 32% |
| | **CYP—Cyclophilin, Peptidyl-Prolyl Cis-Trans Isomerase** | | | | | | | |
| REF2 | ppa002435 | AT3G63400.1 | 61.85 | 11.72 | 91.46 | 47.18 | 1.94 | 19% |
| | **TEF2—LOS1, Translation Elongation Factor** | | | | | | | |
| REF3 | ppa001367 | AT1G56070.1 | 176.3 | 43.46 | 249.4 | 112.9 | 2.21 | 25% |
| REF4 | ppa001368 | AT1G56070.1 | 587.9 | 159.9 | 858.5 | 361.0 | 2.38 | 27% |
| REF5 | ppa020696[b] | AT1G56070.1 | 1.13 | 0.67 | 2.35 | - | - | 59% |
| | **GAPDH Glyceraldehyde-3-Phosphate Dehydrogenase C2** | | | | | | | |
| REF6 | ppa008227 | AT1G13440.1 | 1,523 | 261.2 | 2,112 | 1,079 | 1.96 | 17% |
| | ppa010010 | | 0*** | | | | | |
| | **PLA2—Phospholipase A2-Beta** | | | | | | | |
| REF7 | ppa012884 | AT2G19690.1 | 4.14 | 1.93 | 8.62 | 1.08 | 7.95 | 47% |
| | **RP II-NRPB3, DNA-directed RNA polymerase** | | | | | | | |
| REF8 | ppa008812 | AT2G15430.1 | 115.1 | 33.58 | 169.1 | 51.54 | 3.28 | 29% |
| REF9 | ppa016873 | AT2G15430.1 | 7.77 | 2.09 | 12.15 | 4.05 | 3.00 | 27% |
| | **RPL 13—Ribosomal Protein L13** | | | | | | | |
| REF10 | ppa011512 | AT3G49010.3 | 753.8 | 190.5 | 1,101 | 425.7 | 2.59 | 25% |
| | ppa011516 | | 0[c] | | | | | |
| | ppa011538 | | 0[c] | | | | | |
| | **TUA—Alpha-tubulin5** | | | | | | | |
| REF11 | ppa005642 | AT5G19780.1 | 191.5 | 60.59 | 321.15 | 100.80 | 3.19 | 32% |
| | **TUB—Beta-tubulin1** | | | | | | | |
| REF12 | ppa005644 | AT1G75780.1 | 260.4 | 133.4 | 568.33 | 108.03 | 5.26 | 51% |
| | **UBQ10—Polyubiquitin 10** | | | | | | | |
| REF13 | ppa007117 | AT4G05320.4 | 325.3 | 85.33 | 492.26 | 202.82 | 2.43 | 26% |

[a]Genes identified from Tong et al.,[4]

[b]Expression values too low to be accurate-fewer than 10 reads per library.

[c]not found-no reads were detected for these predicted transcripts.

transcripts was counted and divided by the total number of reads mapped to the peach genome for each of the 20 libraries (S1 Table). Three of the predicted transcripts had no reads that matched and two had less than ten reads per library and were omitted, resulting in 13 reference genes (REF1-REF13). These reference genes were then analyzed utilizing the RNAseq results from 20 different normal and "stoneless" plum libraries (S1 Table) to determine how much variation in expression existed between different tissues at different timepoints. The range of expression values averaged over the 20 RNAseq libraries varied from zero reads per million (RPM) to over 1000 RPM and the standard deviation ranged from 17% to over 50% of the average value. The span within the libraries ranged from ~2 to 8- fold differences, with only REF2 (*CYP)* and REF6 (*GAPDH)* being under 2. In many cases; RPMs from the endocarp tissues (libraries 5,10,15,20, S1 Table) were distinctly different from the early ovary/fruit samples (Fig 1A). Expression levels of reference genes REF1 and REF12 were higher in endocarp, while expression levels of REF3, REF4, REF8, REF9 and REF10 were lower in endocarp when compared to whole ovary/fruit.

Additional previously described reference genes from plum (*Prunus salicina*) and peach (*Prunus persica*) [5–7] were also analyzed in a similar fashion. The results of matching reads

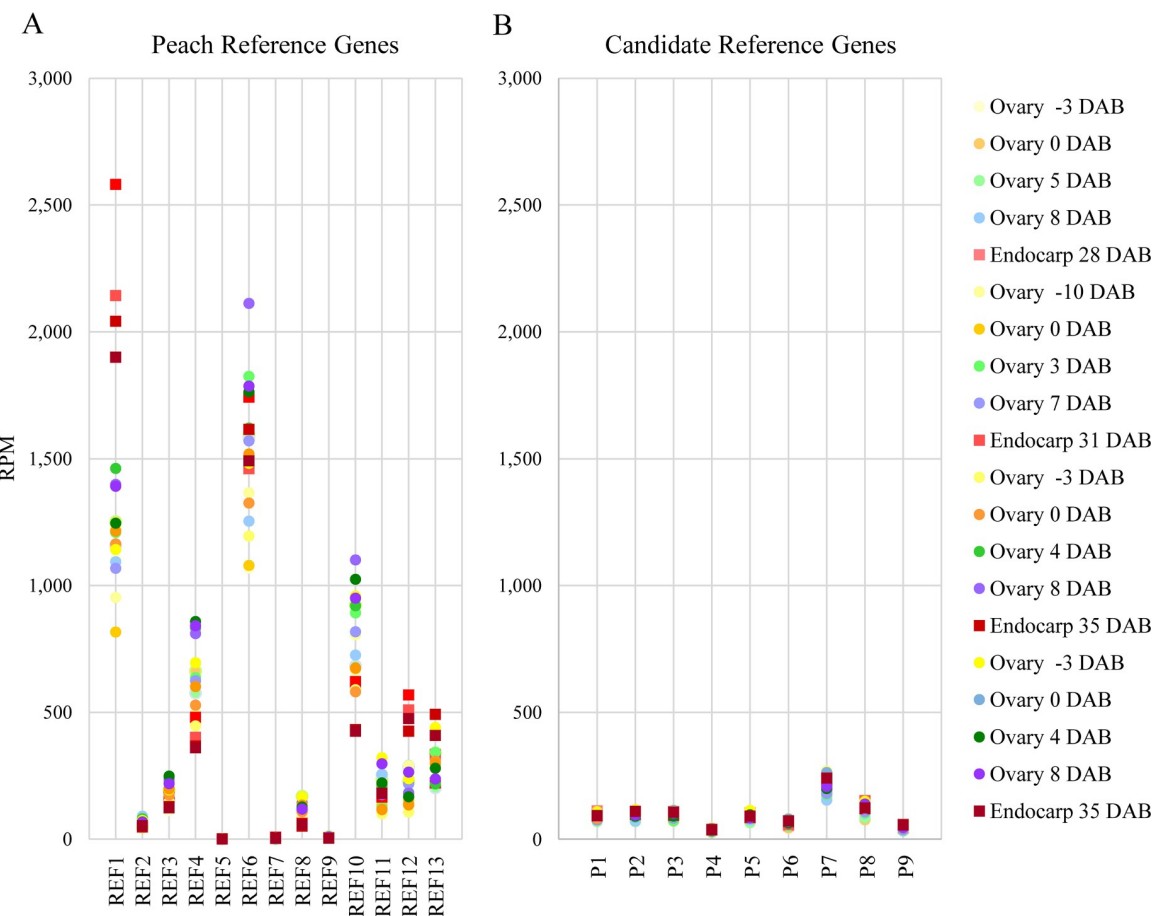

**Fig 1. Comparison of reads per million (RPM) mapped to the peach genome between 13 previously identified reference genes** [4] **and nine candidate reference genes from plum early fruit libraries.** A. Twenty RNAseq libraries were used to determine the RPMs for 13 predicted peach reference genes (REF1-REF13). Each library is represented by a specific color with endocarp tissues represented as squares and fruit/ovary tissues as circles. B. The same twenty RNAseq libraries were analyzed in the same manner as A but for nine candidate reference plum genes (P1-P9).

are shown in S2 Table. In these cases, there were five additional candidates that have less than 2-fold differences in range, *PP2A-2* (ppa009114), *TIP41* (ppa009483), unknown protein (ppa006070), *RPT5A* (ppa006192), and *IQD33* (ppa002870) as well as a CV 20% or less. Most of these genes were identified in an RNAseq set of data and ranged in average RPM in our libraries from 19 to 351.

## Identifying candidate reference genes (P1-P9) in plum early fruit RNAseq libraries

The indications from the RNAseq data were that many of the standard reference genes may not be adequate to normalize gene expression in the early fruit libraries, particularly when comparing endocarp to whole ovary/fruit tissue. The expression patterns of the ~26,000 predicted plum transcripts were then filtered to see if there were more appropriate reference genes for early fruit development. Criteria for new plum reference genes included minimal variation in expression between tissues and time points in normal and the 'Stoneless' cultivars, a minimum expression level above 10 RPM to avoid difficulties in detection, and a maximum expression level below 450 RPM to avoid dilution of the RNA for qPCR.

All genes were first sorted by 0.1-fold changes between 'Stoneless' and the two normal stone cultivars. Those genes that had between |1.0| and |1.1| log2 differences with average normalized read values greater than 11 or less than 450 were kept, resulting in list of 1344 predicted transcripts. These were further sorted by the standard deviation of the CLC (version v5.5) normalized read values for all 20 RNAseq libraries divided by the normalized read value of one 'Stoneless' endocarp library (library 20, S1 Table). This resulted in a value that represented the variation of all the libraries relative to that of an endocarp-based library, giving bias to those genes with less difference in the endocarp. There were 561 predicted transcripts where the standard deviation was less than 20% of the value of the normalized reads from one endocarp library. The previously described reference genes, a *RPT5A* (ppa006192) and an *ELF5A-1* (ppa012654) fell into this group. From this group of 561 genes, nine were chosen from the top 28 genes whose standard deviation (StDev) was less than 11% of the endocarp library values. These were *CorA*-like, *IGPD*, *HAM1*, Ras-related GTP-binding, *pfkB*, a tetratricopeptide repeat containing protein, *SNX1*, *PECT1*, and a *SDH1* related gene, referred to as P1-P9 respectively (Table 2). Neither the *RPT5A* nor the *ELF5A-1* met this criterion. The individual values are plotted in Fig 1B by RPMs for each of the 20 plum libraries used previously,

**Table 2. Candidate reference genes derived from RNAseq plum fruit libraries.**

| Primer Set | Peach Match | Function | *Arabidopsis* Best Match | Mean RPM | StDev | High RPM | Low RPM | High/ Low | CV (StDev/ mean) |
|---|---|---|---|---|---|---|---|---|---|
| P1 | ppa004809 | MRS2-3-magnesium transporter CorA-like | AT3G19640.1 | 64 | 12 | 83 | 44 | 1.89 | 18 |
| P2 | ppa009591 | IGPD-imidazole glcerol-phosphate dehydatase | AT3G22425.2 | 36 | 5 | 46 | 27 | 1.70 | 15 |
| P3 | ppa005747 | HAM1-histone acetyltransferase of the MYST family 1 | AT5G64610.1 | 90 | 11 | 112 | 70 | 1.60 | 12 |
| P4 | ppa017220 | Ras-related GTP-binding | AT5G59840.1 | 121 | 21 | 153 | 77 | 1.99 | 18 |
| P5 | ppa006628 | pfkB-type carbohydrate kinase family protein | AT5G51830.1 | 96 | 12 | 114 | 72 | 1.58 | 12 |
| P6 | ppa004662 | tetratricopeptide repeat containing protein | AT3G15750.1 | 45 | 8 | 59 | 33 | 1.79 | 18 |
| P7 | ppa002552 | SNX1(sorting nexin 1) phosphoino-sititide binding | AT5G06140.1 | 93 | 13 | 117 | 70 | 1.67 | 14 |
| P8 | ppa0056076 | PECT1-phosphorylethanolamine cytidylyltransferase 1 | AT2G38670.1 | 90 | 12 | 114 | 66 | 1.73 | 13 |
| P9 | ppa002787 | SDH1-1-succinate dehydrogenase | AT5G66760.1 | 212 | 33 | 269 | 155 | 1.74 | 15 |

demonstrating that the range of expression values is tighter in the new gene set (P1-P9) than most of the peach reference genes previously reported (REF1-REF13)[4]. The RPMs ranged from 36 to 211 and the largest differences within the libraries ranged from 1.5 to 2-fold. This indicates that from the RNAseq data, a set of genes could be found that is more stable in expression than the standard reference genes, specifically for early fruit development including endocarp tissue.

## qPCR with candidate reference genes

A developmental series (DS) of early fruit RNA from 'Reine Claude de Bavay' beginning with whole flowers (DS1), including ovaries prior to bloom (DS2), and ending with developing whole fruit just prior to stone hardening (DS7), was used to test the candidate reference genes P1-P9 (S3 and S4 Tables). This series of tissues (DS1-DS7) represents the developmental span when endocarp tissue is determined and differentiated up to when it begins to harden. UBQ (S4 Table) was also used as a reference gene. The Cq values are presented in Fig 2. The results were entered in the RefFinder program to determine the comprehensive rank utilizing combined results from several reference gene ranking programs. Most of the normalization programs ranked P2, P3 and P7 at the top, all with stable values from geNorm (0.68, 0.855, and 1.080, respectively) and NormFinder (1.572, 1.224, and 0.538, respectively) (Table 3). These three genes, *IPGD* (imidazole glycerol-phosphate dehydratase), *HAM1* (histone acetyltransferase) and *SNX1* (sorting nexin 1) have not been commonly used as reference genes, but a standard reference gene, UBQ, was always ranked 10th or last in these experiments. The ranking though, did not consider the efficiency of the PCR reactions potentially biasing against those with low efficiency. In the case of our reactions, those with the most divergent efficiencies, P5 (122%) and P7 (117%) were ranked 6th and 1st out of the 10, respectively (S5 Table).

**Fig 2. Cq results from qPCR of nine candidate reference genes and the standard UBQ with a small developmental series of plum early fruit RNAs.** The Cq of each fruit development RNA (average of triplicate samples) is presented for each candidate reference gene. The list of RNAs (DS1-DS7, Stendo1,2 and Cendo1,2) is presented in S3 Table and list of candidate reference genes (P1-P9) is presented in Table 2. qPCRs with the subscript 2 on the X axis represents a new RNA extraction of fruit development that included the endocarp tissues and was assayed with the three most stable of the candidate reference gene as well as UBQ. UBQ represents the *UBIQUITIN10* gene. Primer sets for the reactions are listed in S4 Table.

**Table 3. Results from RefFinder comparing the qPCR profiles for each of the candidate reference genes.**

| Ranking Program | Ranking Order | | | | | | | | | |
|---|---|---|---|---|---|---|---|---|---|---|
| | 1 | 2 | 3 | 4 | 5 | 6 | 7 | 8 | 9 | 10 |
| Recommended Comprehensive ranking | **P7** | **P3** | **P2** | **P1** | **P4** | **P5** | **P8** | **P6** | **P9** | **UBQ** |
| Delta CT[a] | P3 | P7 | P4 | P2 | P1 | P5 | P8 | P6 | P9 | UBQ |
| BestKeeper[b] | P7 | P3 | P1 | P2 | P4 | P5 | P6 | P8 | P9 | UBQ |
| Normfinder[c] | P7 | P4 | P3 | P2 | P1 | P5 | P8 | P6 | P9 | UBQ |
| geNorm[d] | P1 \| P2 | | P3 | P7 | P4 | P5 | P8 | P6 | P9 | UBQ |

[a][15];

[b][16];

[c][17];

[d][18]

## Differential gene expression of test genes (PTs) following normalization with candidate reference genes

In order to test the three top candidate reference genes as well as UBQ, we used them to normalize four "test" genes from plum (PT1-PT4) from the RNAseq early plum fruit and endocarp libraries. These four genes were chosen because they are expressed at different levels in the endocarp tissue of our 'Stoneless' and normal stone cultivars; two had low expression levels overall, PT1 (*HDG11*) and PT3 *(ARA12)* and two had higher expression levels, PT2 (*FER*) and PT4 (*CYP707A1*) (See S4 Table for full names). A second set of RNAs was extracted from the same plum developmental series previously used and four additional RNAs were included, representing two stages of endocarp tissue from both the 'Reine Claude de Bavay' normal stone cultivar (Cendo1 and Cendo2) and from one 'Stoneless' individual (Stendo1 and Stendo2). These were then amplified with the candidate reference genes judged most stable, P2, P3, P7 as well as UBQ and the four endocarp-varying test genes (PT1, PT2, PT3, PT4) (Primers in S4 Table). The comparative results from these four genes are presented without normalization, normalization with UBQ, normalization by the geometric means of all four (P2, P3, P7 and UBQ), the three reference candidate genes (P2, P3, and P7) and without the high value P2 (P3, P7 and UBQ) (Fig 3). They are also presented as normalized by each of the three candidate reference genes individually (S1 Fig). There are only small effects of normalization on the patterns of expression (PT2 and PT4, Fig 3), PT2 peak at DS4 was slightly reduced but more emphasized by the normalization, and for PT4, the rise in expression is more emphasized at DS3 in the normalized data. The other two (PT1 and PT3) do show more dramatic differences when normalized. For example, PT1 has relatively equal amounts for DS3, DS4 and DS5, but when standardized with UBQ as well as combined standard genes has a much larger increase at DS3 which then drops. The difference between the normal endocarp tissue and 'Stoneless' endocarp for both PT1 and PT3 is also much greater when standardized with any of the genes or combinations. Neither of these genes would have been chosen as varying between normal and 'Stoneless' endocarp tissue from qPCR without the normalization.

## Universality of the early fruit candidate reference genes

The usefulness of these early fruit reference genes was tested in two other Rosaceae species, peach (*Prunus persica*) and apple (*Malus domestica*). We looked in existing RNAseq data sets to determine if these might be good candidates, much as we had done originally with the published *Prunus* reference genes. In this case, we used three different RNAseq experiments, one that looked at four different fruit tissues of peach at four time points, from 0 days after anthesis

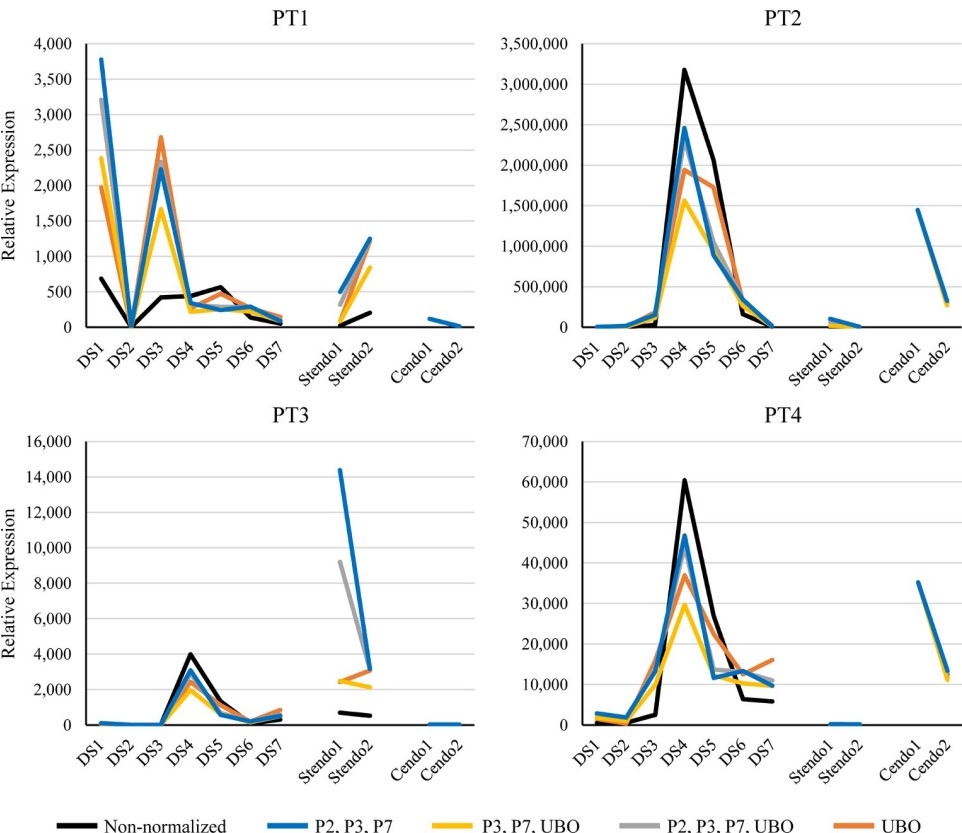

**Fig 3. Normalization of expression data for four genes in a plum early fruit series and in "Stoneless" and normal endocarp tissue.** Each line represents a different combination of reference genes used to normalize the values. The black line represents the non-normalized values. The DS1-DS7 and endocarp RNA samples, are listed in S3 Table and the primer sets and gene identity for PT1, PT2, PT3 and PT4 are listed in S4 Table. Stendo = 'Stoneless' endocarp. Cendo = 'Reine Claude de Bavay' endocarp.

(DAA) to 18 DAA. The sample libraries were in triplicate. The second set was in apple and also looked at four different fruit tissues at four different stages from 0 DAA to 20 DAA. The sample libraries were in triplicate. The last set was also in apple and looked at the effects of two hormone treatments on three different fruit tissue at 18 DAA and at 132 DAA as well as a no pollination control and a hand pollinated control, all in triplicate samples [19]. The nine reference genes identified in plum had already been compared to peach to identify orthologs. For

**Table 4. Overall coefficient of variance for RNAseq experiments in early Peach and Apple fruit for nine candidate reference genes.**

| | **P1** | | **P2** | | **P3** | | **P4** | | **P5** | | **P6** | | **P7** | | **P8** | | **P9** | |
|---|---|---|---|---|---|---|---|---|---|---|---|---|---|---|---|---|---|---|---|
| **Peach Development** | | | | | | | | | | | | | | | | | | | |
| Transcript | PeD | | PeD | | PeD | | PeD | | PeD | | PeD | | PeD | | PeD | | PeD | |
| All Libraries | 38 | | 17 | | 23 | | 20 | | 19 | | 17 | | 15 | | 22 | | 39 | |
| **Apple Development** | | | | | | | | | | | | | | | | | | | |
| Transcript | AD 1 | AD 2 | AD3 | AD 4 | AD5 | AD6 | AD 7 | AD 8 | AD 9 | AD10 | AD 11 | AD 12 | AD13 | AD 14 | AD15 | AD 16 | AD 17 | AD 18 |
| All Libraries | 24 | 24 | 18 | 15 | 27 | 28 | 14 | 19 | 152 | 45 | 34 | 21 | 21 | 13 | 28 | 16 | 45 | 61 |
| **Apple Hormone Treated** | | | | | | | | | | | | | | | | | | | |
| Transcript | AH 1 | AH 2 | AH3 | AH 4 | AH5 | AH6 | AH 7 | AH8 | AH 9 | AH10 | AH 11 | AH 12 | AH13 | AH14 | AH15 | AH 16 | AH 17 | AH 18 |
| All Libraries | 31 | 25 | 45 | 29 | 38 | 42 | 37 | 30 | 44 | 61 | 51 | 34 | 14 | 16 | 31 | 38 | 45 | 42 |

apple they were used to identify the 2 closest related genes because of the duplicated apple genome (S6 Table). The coefficient of variance (STD/mean) (CV) for each of the three experiments are presented as a comparison of all the libraries in each experiment (Table 4). A breakdown of each experiment to look at where the most variation and stability was done to further understand the usefulness of these candidate reference genes (S7 Table).

In the peach development and tissue series, most of the comparisons had a 20% or less CV. Notable exceptions were P1 (PeD1) and P9 (PeD9) which had 38% and 39% respectively for a comparison of all 48 libraries (Table 4). P2 (PeD2) and P7 (PeD7) had less than 20%CV for all comparisons (S7 Table).

Apple had even more variation for the reference genes in both the development and the hormone treatment data sets. In the apple development and tissue series, both P2 orthologs (AD3, AD4) as well as P4 ortholog (AD7), P7 ortholog (AD14), P8 ortholog (AD16) had less than 20%CV for all library comparisons (Table 4). But for both P5 orthologs (AD9, AD10) and P9 orthologs (AD17, AD18) as well as one of the P6 orthologs (AD11) CV values exceeding 50% for some of the library comparisons (S7 Table).

For the hormone treated sets, only the P7 orthologs had values for all libraries below 20% and within each group the CV was below 20% (Tables 4 and S7). All the other reference genes had at least one comparison that was above 38% with again the P5 orthologs having the highest CVs. When all the reference genes were compared with one set of libraries, the only consistency was with the tissues that were senescing, the GA treated ovule, all the NAA treated tissues and the NEG control ovary wall. This lack of variation was mostly due to consistently low expression (S7 Table).

## Discussion

Expression profiling of specific genes is an important approach in determining the correlation and, therefore, the potential involvement of genes in the processes under exploration. Utilizing qPCR has been an accepted technique for quantifying RNA expression as well as verifying other means of quantifying RNA expression like RNAseq experiments [20,21]. However, the reliability of the qPCR quantification is dependent on having a reference gene to compare its expression with the experimental gene expression [22,23]. If the reference gene varies in expression, the conclusion from the experimental genes may be erroneous [24,25].

We are interested in early fruit development, in particular the formation of the endocarp, or stone tissue, in stone fruit. We used an RNAseq approach to investigate the gene expression blanketing the time of endocarp differentiation in three different cultivars of plum, two which had normal endocarp, and one exhibiting reduced endocarp. To confirm the RNAseq results qPCR has been selected, highlighting the need of a set of reference genes from endocarp and early fruit developmental stages to help us standardize RNA expression. We identified standard genes from the literature that had been reported for stone [4–7] and used those to extract the RPM values from our 20 expression libraries representing early plum fruit development and young endocarp tissue. Only a few of these genes appeared to be stably expressed in all the libraries which prompted us to utilize the RNAseq data to generate a specific set of reference genes (Tables 1 and S2). Genes expressed in the early plum fruit libraries and early endocarp tissue libraries that varied no more than 10% amongst the libraries, and with a minimum and maximum expression level not greater than 2-fold, were selected and nine of those genes further tested by qPCR (Table 2). Three of the candidate reference genes, P2, P3 and P7, *IPGD* (imidazole glycerol-phosphate dehydratase), *HAM1* (histone acetyltransferase) and *SNX1* (sorting nexin 1) respectively, had relatively similar expression in all the fruit development samples as well as endocarp tissues tested (Figs 2 and 3). These three genes, along with a

previous standard UBQ, were then used to standardize the expression of four chosen genes exhibiting differential expression in the endocarp tissues of normal stone cultivar compared to 'Stoneless'. Those differences were detected in the qPCR for two of the genes (PT2, PT415) regardless of standardizing the expression, while the other two genes (PT1, PT3) showed differential expression when standardized, emphasizing the importance of reference genes.

The nine genes were further looked at as potential reference genes for two other species, apple and peach, using both early fruit developmental series and different fruit tissue types, as well as growth regulator treated fruit. Once again, the expression of three genes were relatively stable in the fruit development series though they were not as stable in the hormone treated experiments in apple. One gene, P7, however was extremely stable in all tissues, all times and also under hormone treatments. This gene is a sorting nexin 1 which is part of a complex of proteins involved in trafficking proteins [26].

The use of reference genes to standardize expression of other genes has evolved from a set chosen thought to be constitutive or 'housekeeping' [4,27] to highly specific sets tailored to varying conditions and tissue types [28,29]. With the advent of new technologies, there are many data sets available of expressed RNAs specific to various tissues, developmental times and experimental tissues. These can then be mined for genes that are better suited to represent the unvarying gene expression control for specific experiments. The demonstration here for early fruit tissues of plum further underlines the fruitfulness of such an approach rather than rely on 'housekeeping' genes.

## Conclusions

The majority of the previously published *Prunus* reference genes for RNA expression had variable expression in early plum (*Prunus domestica*) fruit, especially in the endocarp tissue relative to the whole fruit tissue. Nine candidate reference genes were chosen from a set of RNAseq data from early plum fruit and endocarp tissue, that were more stable and tested in a plum fruit developmental series. The three best genes as defined using RefFinder were used to normalize expression of a plum fruit developmental series and endocarp tissues from a normal endocarp plum and endocarp tissue from an endocarp mutant. Normalization with any of those three as well as a combination, resulted in the expected difference in endocarp expression in the mutant endocarp. One of the candidate reference genes, SNX1 appears to be a universal reference gene for Rosaceae fruit tissues.

## Supporting information

**S1 Fig. Normalization of expression data for four genes in a plum early fruit series and in "Stoneless" and normal endocarp tissue.** Each line represents a different single reference gene used to normalize the values. The black line represents the non-normalized values. The DS1-DS7 and endocarp RNA samples are listed in S3 Table and the primer sets and gene identity for PT1, PT2, PT3 and PT4 are listed in S4 Table. Stendo = 'Stoneless" endocarp. Cendo = 'Reine Claude de Bavay' endocarp.
(TIF)

**S1 Table. Plum RNA libraries used for RNAseq.**
(DOCX)

**S2 Table. Expression values of Prunus genes in RNAseq libraries.**
(DOCX)

**S3 Table. Developmental series of plum fruit and endocarp tissues.**
(DOCX)

**S4 Table. Primer pairs for reference genes and endocarp varying genes.**
(DOCX)

**S5 Table. Reference candidate gene descriptions.**
(DOCX)

**S6 Table. Peach and Apple orthologs.**
(DOCX)

**S7 Table. Coefficient of variance for RNAseq experiments in early Peach and Apple fruit for the nine candidate reference genes.**
(DOCX)

## Acknowledgments

The authors would like to acknowledge the contribution of Linda Dunn and Dan Bullock for assistance in the design and testing of the primers, Kyle Ebersole for some of the qPCR studies, and Mark Demuth and Larry Crim for managing the plum, peach and apple trees from which fruit were harvested.

## Author Contributions

**Conceptualization:** Kelsey Galimba, Roberta Tosetti, Ann Callahan.

**Data curation:** Ann Callahan.

**Formal analysis:** Kelsey Galimba, Roberta Tosetti, Karen Loerich, Leann Michael, Savita Pabhakar, Ann Callahan.

**Funding acquisition:** Chris Dardick, Ann Callahan.

**Investigation:** Kelsey Galimba, Karen Loerich, Leann Michael, Savita Pabhakar, Ann Callahan.

**Project administration:** Cynthia Dove, Ann Callahan.

**Supervision:** Savita Pabhakar, Cynthia Dove, Chris Dardick.

**Writing – original draft:** Ann Callahan.

**Writing – review & editing:** Kelsey Galimba, Roberta Tosetti, Karen Loerich, Leann Michael, Savita Pabhakar, Cynthia Dove, Chris Dardick, Ann Callahan.

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
