## [Decision Letter · Decision Letter 0]

12 Nov 2019

PONE-D-19-27213

Identification of early fruit development reference genes in plum

PLOS ONE

Dear Dr. Callahan,

Thank you for submitting your manuscript to PLOS ONE. After careful consideration, we feel that it has merit but does not fully meet PLOS ONE’s publication criteria as it currently stands. Therefore, we invite you to submit a revised version of the manuscript that addresses the points raised during the review process.

i) The presentation needs serious improvements.

We would appreciate receiving your revised manuscript by Dec 27 2019 11:59PM. To enhance the reproducibility of your results, we recommend that if applicable you deposit your laboratory protocols in protocols.io, where a protocol can be assigned its own identifier (DOI) such that it can be cited independently in the future. For instructions see: http://journals.plos.org/plosone/s/submission-guidelines#loc-laboratory-protocols

We look forward to receiving your revised manuscript.

Kind regards,

Yun Zheng, Ph.D

Academic Editor

PLOS ONE

Journal Requirements:

1. We note that you have stated that you will provide repository information for your data at acceptance. Should your manuscript be accepted for publication, we will hold it until you provide the relevant accession numbers or DOIs necessary to access your data. If you wish to make changes to your Data Availability statement, please describe these changes in your cover letter and we will update your Data Availability statement to reflect the information you provide.

Reviewers' comments:

Reviewer's Responses to Questions

**Comments to the Author**

1. Is the manuscript technically sound, and do the data support the conclusions?

Reviewer #1: Yes

Reviewer #2: Yes

2. Has the statistical analysis been performed appropriately and rigorously? 

Reviewer #1: Yes

Reviewer #2: No

3. Have the authors made all data underlying the findings in their manuscript fully available?

Reviewer #1: Yes

Reviewer #2: Yes

4. Is the manuscript presented in an intelligible fashion and written in standard English?

Reviewer #1: Yes

Reviewer #2: Yes

5. Review Comments to the Author

Reviewer #1: In this manuscript, the authors tried to identify sets of genes that could be used to normalize expression studies in early fruit development, using RNAseq/qPCR experiments in a series of developmental plum fruit samples. They found nine newly identified candidate reference genes that appeared to be more consistent in the RNAseq data from early fruit libraries than the previously identified Prunus reference genes. Their results suggest that the re-mining of existing RNAseq data can find novel reference genes for a specific tissues or experimental conditions. I think this work is interesting and resourceful, the experiments and statistical analysis used meet the standard, and the manuscript is acceptable for publication.

Reviewer #2: This study used existing RNAseq data to identify standard genes to provide stable reference genes for early fruit development and endocarp tissue in plum. Nine candidate reference genes were chosen from a set of RNAseq data, three best genes as defined using RefFinder were used to normalize expression of a plum fruit developmental series and endocarp tissues from a normal endocarp plum and endocarp tissue from an endocarp mutant. Normalization with any of those three as well as a combination resulted in difference in endocarp expression in the mutant endocarp. It is an interesting study, however there still some concerns with the data.

1. The candidate reference gene P4 looks even stable between different library than P1 and P2 in figure 1, why don’t you choose it?

2. Figure 1, the X axis legend don’t fit well with the Y axis data which make confusion.

3. Figure 3, the black line represents the non-normalized values, but the legend is P10. The Blue and grey line missed the legend. The X axis legend for P10 and p11 are partly missing. Please make sure the figures are presented scientifically and rigorously.

6. PLOS authors have the option to publish the peer review history of their article (what does this mean?). If published, this will include your full peer review and any attached files.

Reviewer #1: No

Reviewer #2: No

---

## [Author Response · Author response to Decision Letter 0]

15 Jan 2020

Response to Reviewers Comments:

Reviewer #1: In this manuscript, the authors tried to identify sets of genes that could be used to normalize expression studies in early fruit development, using RNAseq/qPCR experiments in a series of developmental plum fruit samples. They found nine newly identified candidate reference genes that appeared to be more consistent in the RNAseq data from early fruit libraries than the previously identified Prunus reference genes. Their results suggest that the re-mining of existing RNAseq data can find novel reference genes for a specific tissues or experimental conditions. I think this work is interesting and resourceful, the experiments and statistical analysis used meet the standard, and the manuscript is acceptable for publication.

Reviewer #2: This study used existing RNAseq data to identify standard genes to provide stable reference genes for early fruit development and endocarp tissue in plum. Nine candidate reference genes were chosen from a set of RNAseq data, three best genes as defined using RefFinder were used to normalize expression of a plum fruit developmental series and endocarp tissues from a normal endocarp plum and endocarp tissue from an endocarp mutant. Normalization with any of those three as well as a combination resulted in difference in endocarp expression in the mutant endocarp. It is an interesting study, however there still some concerns with the data.

1. The candidate reference gene P4 looks even stable between different library than P1 and P2 in figure 1, why don’t you choose it?

Candidate Reference Gene P4 appears to have a compact distribution spread and good efficiency but was ranked 5th in the Recommended Comprehensive ranking in RefFinder, with no 1st place rankings in any of the programs. We focused on the top three genes in the Recommended Comprehensive ranking, which are all 1st in at least one program. We added this to line 277 that we only proceeded with the top three genes and UBQ for the qPCR normalization experiments.

2. Figure 1, the X axis legend don’t fit well with the Y axis data which make confusion.

We have re-labeled the x-axis with the appropriate gene IDs. We have also changed the colors so that similar tissues are similar colors, and moved the legend so that this figure is more clear. 

3. Figure 3, the black line represents the non-normalized values, but the legend is P10. The Blue and grey line missed the legend. The X axis legend for P10 and p11 are partly missing. Please make sure the figures are presented scientifically and rigorously.

We thank the reviewer for bringing this to our attention - the formatting of this figure was altered when it was uploaded. We have fixed the issues with the missing lines. We have changed the black line to “non-normalized” in the legend and we have simplified the legend to be shared between all four graphs. We also corrected S1 Fig which had similar issues.

1. Is the manuscript technically sound, and do the data support the conclusions?

Reviewer #1: Yes

Reviewer #2: Yes

2. Has the statistical analysis been performed appropriately and rigorously? 

Reviewer #1: Yes

Reviewer #2: No

We hope that this has been clarified in the more correct presentation of the graphs. We reworded some of the results section to present a clearer picture of what we were doing. (lines 208-211; 217-227; 234-241) as well as retitling figure legend 3.

3. Have the authors made all data underlying the findings in their manuscript fully available?

Reviewer #1: Yes

Reviewer #2: Yes

We are still in the process of submitting all the sequence data sets to SRA (NCBI). One set is completed but the other 4 are in the process. We will submit the accession numbers when we obtain all of them.

4. Is the manuscript presented in an intelligible fashion and written in standard English?

Reviewer #1: Yes

Reviewer #2: Yes

In addition to the suggested edits, we also made some cosmetic changes in order to improve the clarity of the manuscript. We thought this would help with the data interpretations.

1. We changed some of the abbreviations we assigned to genes. 

Previously published reference genes are now referred to as REF1-REF13 instead of S1-S13. Candidate reference genes from plum are still referred to as P1-P9. The four plum genes that were used to test the normalization with P1-P9 are now referred to as “plum test” PT1-PT4 instead of P10-P15. Peach orthologs used to test normalization in the peach developmental series are now referred to as PeD1-PeD9. Apple orthologs from version 1 of the apple genome that are used to normalize the apple developmental series are now referred to as AD1-AD18 instead of D1-D18. Apple orthologs from the GDDH13 version of the apple genome used to normalize the apple hormone samples are now referred to as AH1-AH18 instead of H1-18. 

2. We included frequent references to these abbreviations throughout the text in order to make it easier for the reader to keep track of them. 

3. We changed the wording of some sentences/paragraphs to make them easier to understand.

---

## [Decision Letter · Decision Letter 1]

12 Mar 2020

Identification of early fruit development reference genes in plum

PONE-D-19-27213R1

Dear Dr. Callahan,

We are pleased to inform you that your manuscript has been judged scientifically suitable for publication and will be formally accepted for publication once it complies with all outstanding technical requirements.

With kind regards,

Yun Zheng, Ph.D

Academic Editor

PLOS ONE

Additional Editor Comments (optional):

Reviewers' comments:

Reviewer's Responses to Questions

**Comments to the Author**

1. If the authors have adequately addressed your comments raised in a previous round of review and you feel that this manuscript is now acceptable for publication, you may indicate that here to bypass the “Comments to the Author” section, enter your conflict of interest statement in the “Confidential to Editor” section, and submit your "Accept" recommendation.

Reviewer #1: All comments have been addressed

Reviewer #2: All comments have been addressed

2. Is the manuscript technically sound, and do the data support the conclusions?

Reviewer #1: Yes

Reviewer #2: Yes

3. Has the statistical analysis been performed appropriately and rigorously? 

Reviewer #1: Yes

Reviewer #2: Yes

4. Have the authors made all data underlying the findings in their manuscript fully available?

Reviewer #1: Yes

Reviewer #2: Yes

5. Is the manuscript presented in an intelligible fashion and written in standard English?

Reviewer #1: Yes

Reviewer #2: Yes

6. Review Comments to the Author

Reviewer #1: (No Response)

Reviewer #2: (No Response)

7. PLOS authors have the option to publish the peer review history of their article (what does this mean?). If published, this will include your full peer review and any attached files.

Reviewer #1: No

Reviewer #2: No

---

## [Editor Report · Acceptance letter]

27 Mar 2020

PONE-D-19-27213R1 

Identification of early fruit development reference genes in plum 

Dear Dr. Callahan:

I am pleased to inform you that your manuscript has been deemed suitable for publication in PLOS ONE. Congratulations! Your manuscript is now with our production department. 

With kind regards,

on behalf of

Dr. Yun Zheng 

Academic Editor

PLOS ONE